# Two-Stage Diffusion Models: Better Image Synthesis by Explicitly Modeling Semantics

## Abstract

Recent progress with conditional image diffusion models has been stunning, and this holds true whether we are speaking about models conditioned on a text description, a scene layout, or a sketch. Unconditional image diffusion models are also improving but lag behind, as do diffusion models which are conditioned on lower-dimensional features like class labels. We advocate for a simple method that leverages this phenomenon for better unconditional generative modeling. In particular, we suggest a two-stage sampling procedure. In the first stage we sample an embedding describing the semantic content of the image. In the second stage we use a conditional image diffusion model to sample the image conditioned on this embedding, and then discard the embedding. The combined model can therefore leverage the power of conditional diffusion models on the unconditional generation task, achieving large improvements in unconditional image generation. The same method can be generalized to yield similar improvements for image generation conditioned on a low-dimensional signal like a class label.

## 1 Introduction

Recent text-to-image diffusion generative models (DGMs) have exhibited stunning sample quality (Saharia et al., 2022) to the point that they are now being used to create art (Oppenlaender, 2022). Further work has explored conditioning on scene layouts (Zhang & Agrawala, 2023), segmentation masks (Zhang & Agrawala, 2023; Hu et al., 2022), or the appearance of a particular object (Ma et al., 2023). We broadly lump these methods together as "conditional" DGMs to contrast them with "unconditional" image DGMs which sample an image without dependence on text or any other information. Relative to unconditional DGMs, conditional DGMs typically produce more realistic samples (Ho & Salimans, 2022; Bao et al., 2022; Hu et al., 2022) and work better with few sampling steps (Meng et al., 2022). Furthermore our results suggest that sample realism grows with "how much" information the DGM is conditioned on. We therefore distinguish between "strongly-conditional" generation, where we condition on a high-dimensional feature like a long text prompt, and "lightly-conditional" generation, where

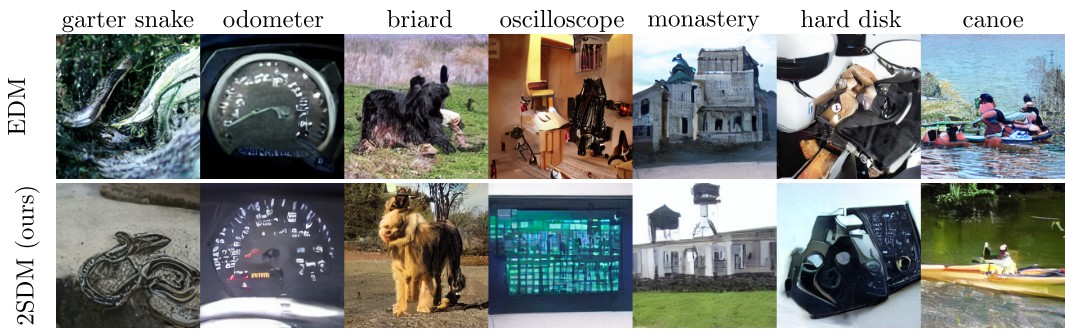

Figure 1: Class-conditional ImageNet-256 samples from our method, 2SDM, and a diffusion model baseline, EDM (Karras et al., 2022), both trained for 12 GPU days. Samples within the same column are generated with the same random seed and class label. In most columns the samples from 2SDM are visibly better, agreeing with the FIDs reported in Section 5.

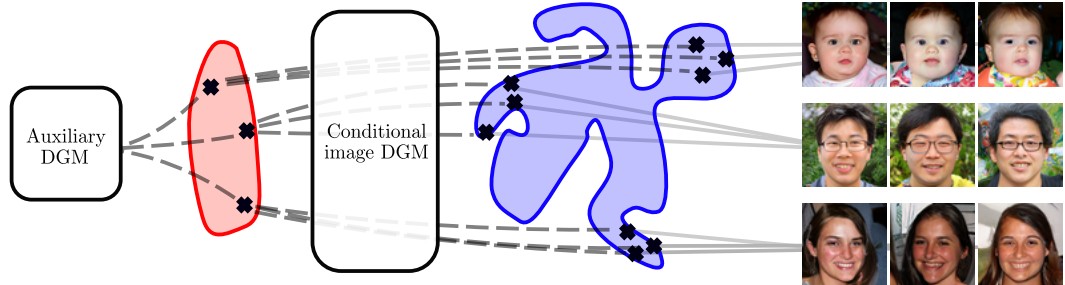

Figure 2: Visualization of 2SDM's generation process. First the auxiliary DGM samples a CLIP embedding, corresponding to a cross in the space of CLIP embeddings (red) in our illustration. Next, our conditional image model maps from the sampled CLIP embedding to a sampled image, visualized on the image manifold (blue). The distribution over plausible images is complex and multi-modal but becomes simpler when conditioned on a CLIP embedding. On the right we show three rows of sampled images. Within each row, all images are generated given the same CLIP embedding.

we condition on a lower dimensional feature like a class label or short text prompt. As hinted at in Fig. 3 an image is likely to be more realistic if conditioned on being "an aerial photograph of a road between green fields" (strongly-conditional generation) than if it is if simply conditioned on being "an aerial photograph" (lightly-conditional generation).

This gap in performance is problematic. Imagine you need to sample a dataset of synthetic aerial photos.[2] A researcher doing so would currently have to either (a) make up a scene description before generating each dataset image, and ensure these cover the entirety of the desired distribution, or (b) accept the inferior image quality gleaned by conditioning just on

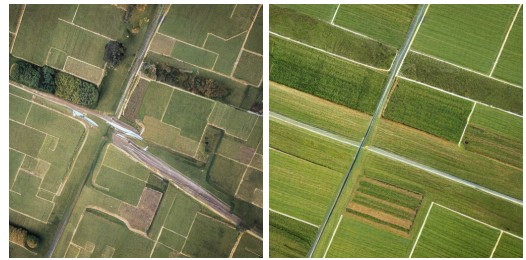

Figure 3: **Left:** Output from Stable Diffusion (Rombach et al., 2022) prompted to produce "aerial photography". **Right:** Using a more detailed prompt[1] with the same random seed removes the "smudged" road artifact that appears on the left. 2SDM builds on this observation.

each image being "an aerial photograph". Figure 3 shows that the difference in quality can be stark.

We argue that a solution to this problem comes from revisiting the methodology of DALL-E 2, also known as unCLIP (Ramesh et al., 2022). UnCLIP is a method for text-conditional image generation which we describe in detail in Section 2. It was originally proposed as a way to "invert" a pretrained CLIP embedder and thereby map from text to image space but, perhaps due to improved text embeddings and a desire for methodological simplicity, we are not aware of future work building on the two-stage unCLIP approach (Rombach et al., 2022; Chang et al., 2023; Hoogeboom et al., 2023). We hope to counter this trend, arguing that, while unCLIP may provide little benefit for "strongly-conditional" text-to-image generation (especially when the text prompt is long or heavily "prompt-engineered"), its benefits are in fact much greater than previously acknowledged when applied to unconditional or "lightly-conditional" generation.

Our final approach, based on unCLIP, is depicted in Fig. 2. A first "auxiliary DGM" samples vectors within an embedding space, with any vector describing a particular set of semantic characteristics of an image. The second stage, a "conditional image DGM", takes such a vector as input and samples an image with these semantic characteristics. The vector embedding is informative, as evidenced by the fact that all images within each row on the right of Fig. 2, which are all conditioned on the same embedding, look very similar. The conditional image DGM therefore inherits all the previously-described advantages of strongly-conditional DGMs even though our overall generative model is

---

[1]We used the prompt "Aerial photography of a patchwork of small green fields separated by brown dirt tracks between them. A large tarmac road passes through the scene from left to right."

[2]This may be done to, e.g., later train state-of-the-art a classification system (Azizi et al., 2023).

unconditional (or, with the generalization in Section 4, lightly-conditional). We call the resulting model a Two-Stage Diffusion Model (2SDM).

**Contributions** In Sections 2 and 3 we revisit unCLIP and then provide a novel explanation for why it is well-suited to the unconditional and lightly-conditional setting which was not explored by Ramesh et al. (2022). We then demonstrate empirically that our lightly-conditional variant, 2SDM, yields large improvements on a variety of image datasets, tasks, and metrics in Section 5.

## 2 BACKGROUND

**Conditional DGMs** We provide a high-level overview of conditional DGMs that is sufficient to understand our contributions, referring to Karras et al. for a more complete description and derivation. A conditional image DGM (Tashiro et al., 2021) samples an image $\mathbf{x}$ given a conditioning input $\mathbf{y}$, where $\mathbf{y}$ can be, for example, a class label, a text description, or both of these in a tuple. We can recover an unconditional DGM by setting $\mathbf{y}$ to a null variable in the below. Given a dataset of $(\mathbf{x}, \mathbf{y})$ pairs sampled from $p_{\text{data}}(\cdot, \cdot)$, a conditional DGM $p_\theta(\mathbf{x}|\mathbf{y})$ is fit to approximate $p_{\text{data}}(\mathbf{x}|\mathbf{y})$. It is parameterized by a neural network $\hat{\mathbf{x}}_\theta(\cdot)$ trained to optimize

$$\mathbb{E}_{u(\sigma)p_\sigma(\mathbf{x}_\sigma|\mathbf{x},\sigma)p_{\text{data}}(\mathbf{x},\mathbf{y})} \left[ \lambda(\sigma)||\mathbf{x} - \hat{\mathbf{x}}_\theta(\mathbf{x}_\sigma, \mathbf{y}, \sigma)||^2 \right] \tag{1}$$

where $\mathbf{x}_\sigma \sim p_\sigma(\cdot|\mathbf{x}, \sigma)$ is a copy of $\mathbf{x}$ corrupted by Gaussian noise with standard deviation $\sigma$; $u(\sigma)$ is a broad distribution over noise standard deviations; and $\lambda(\sigma)$ is a weighting function. If $\lambda$ and $u$ are chosen appropriately, Eq. (1) is a lower bound on the data likelihood. It is common to instead set $\lambda$ and $u$ to values that maximize perceptual quality of the generated images but there remains a close relationship to the ELBO (Kingma & Gao, 2023). During inference, samples from $p_\theta(\mathbf{x}|\mathbf{y})$ are drawn via a stochastic differential equation with dynamics dependent on $\hat{\mathbf{x}}_\theta(\cdot)$.

**CLIP embeddings** CLIP (contrastive language-image pre-training) (Radford et al., 2021) consists of two neural networks, an image embedder $e_i(\cdot)$ and a text embedder $e_t(\cdot)$, trained on a large captioned-image dataset. Given an image $\mathbf{x}$ and a caption $\mathbf{y}$, the training objective encourages the cosine similarity between $e_i(\mathbf{x})$ and $e_t(\mathbf{y})$ to be large if $\mathbf{x}$ and $\mathbf{y}$ are a matching image-caption pair and small if not. The image embedder therefore learns to map from an image to a semantically-meaningful embedding capturing any features that may be included in a caption. We use a CLIP image embedder with the ViT-B/32 architecture and weights released by Radford et al. (2021). We can visualize the information captured by the CLIP embedding by showing the distribution of images produced by our conditional DGM given a single CLIP embedding; see Fig. 2.

**UnCLIP for text-to-image** UnCLIP (Ramesh et al., 2022) uses the following text-to-image procedure: given a text prompt, it is embedded by a CLIP text embedder. A diffusion model then samples a plausible CLIP image embedding with high cosine similarity to this text image embedding. Finally, a conditional image diffusion model samples an image conditioned on CLIP image embedding and text prompt. This is described as "inverting" the CLIP embedder framework to map from image to text, hence the name unCLIP. In the next section we investigate when and why the quality of images produced by a CLIP-conditional image DGM may be greater than those generated by an unconditional image DGM.

## 3 CONDITIONAL VS. UNCONDITIONAL DGMS

**What does it mean to say that conditional DGMs beat unconditional DGMs?** A standard procedure to evaluate unconditional DGMs is to start by sampling a set of $N$ images independently from the model: $\mathbf{x}^{(1)}, \ldots, \mathbf{x}^{(N)} \sim p_\theta(\cdot)$. We can then compute the Fréchet Inception distance (FID) (Heusel et al., 2017) between this set and the dataset. If the generative model matches the data distribution well, the FID will be low. For conditional DGMs the standard procedure has one extra step: we first independently sample $\mathbf{y}^{(1)}, \ldots, \mathbf{y}^{(N)} \sim p_{\text{data}}(\cdot)$. We then sample each image given the corresponding $\mathbf{y}^{(i)}$ as $\mathbf{x}^{(i)} \sim p_\theta(\cdot|\mathbf{y}^{(i)})$. Then, as in the unconditional case, we compute the FID between the set of images $\mathbf{x}_1, \ldots, \mathbf{x}_N$ and the dataset, without reference to $\mathbf{y}_1, \ldots, \mathbf{y}_N$. Even though it does not measure alignment between $\mathbf{x}, \mathbf{y}$ pairs, conditional DGMs beat comparable

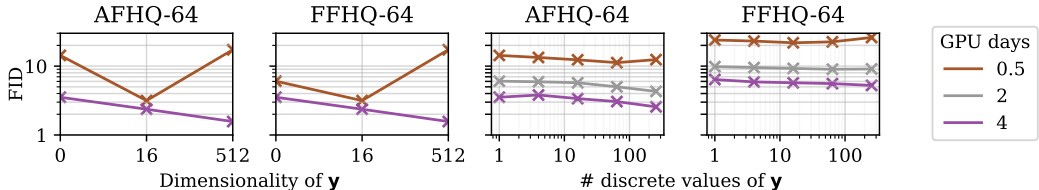

Figure 4: FID versus dimensionality of $\mathbf{y}$ on AFHQ (Choi et al., 2020) and FFHQ (Karras et al., 2018). With small training budgets (brown line), it is harmful when $\mathbf{y}$ is too informative. With larger training budgets (purple line), it is helpful to make $\mathbf{y}$ much more high dimensional.

unconditional DGMs on this metric in many settings: class-conditional CIFAR-10 generation (Karras et al., 2022), segmentation-conditional generation (Hu et al., 2022), or bounding box-conditional generation (Hu et al., 2022).

**Why do conditional DGMs beat unconditional DGMs?**   Conditional DGMS "see" more data during training than their unconditional counterparts because updates involve $\mathbf{y}$ as well as $\mathbf{x}$. Bao et al. (2022); Hu et al. (2022) prove that this is not the sole reason for their successes because the effect holds up even when $\mathbf{y}$ is derived from an unconditional dataset through self-supervised learning. To our knowledge, the best explanation for their success is, as stated by Bao et al. (2022), that conditional distributions typically have "fewer modes and [are] easier to fit than the original data distribution."

**When do conditional DGMs beat unconditional DGMS?**   We present results in Fig. 4 to answer this question. We show FID scores for conditional DGMs trained to condition on embeddings of varying information content. We produce $\mathbf{y}$ by starting from the CLIP embedding of each image in our dataset and using either principal component analysis to reduce their dimensionality (left two panels) or K-means clustering to discretize them (right two panels) (Hu et al., 2022). We see that, given a small training budget, it is best to condition on little information. With a larger training budget, performance appears to improve steadily as the dimensionality of $\mathbf{y}$ is expanded. We hypothesize that **(1)** conditioning on higher-dimensional $\mathbf{y}$ slows down training because it means that points close to any given value of $\mathbf{y}$ will be seen less frequently and **(2)** with a large enough compute budget, any $\mathbf{y}$ correlated with $\mathbf{x}$ will be useful to condition on. This suggests that, as compute budgets grow, making unconditional DGM performance match conditional DGM performance will be increasingly useful.

**A perspective on unCLIP**   Recall that unCLIP leverages a CLIP-conditional generative model even when the original task calls for only a text-conditional image generative model. In light of this section, it makes sense that this should provide a benefit as long as the combination of text and CLIP embedding contains "more" information than the text prompt alone, which will always be the case. However, the disparity is even larger if we compare the CLIP-conditional generative model with an unconditional generative model (i.e. one conditioned on zero bits of information). The unCLIP approach can therefore be expected to provide larger benefits for unconditional (or lightly-conditional) generation than for the text-conditional setting in which it was developed.

## 4   METHOD

We now formally introduce 2SDM, a variant of unCLIP for the unconditional setting. Recall that, for unconditional generation, the user does not wish to specify any input to condition on and, for the lightly-conditional setting, any such input is low-dimensional. We will denote any input $\mathbf{a}$ (letting $\mathbf{a}$ be a null variable in the unconditional setting) and from now on always use $\mathbf{y} := e_i(\mathbf{x})$ to refer to a CLIP embedding. To make this deterministic encoding compatible with a probabilistic generative modeling perspective, we consider a joint distribution $p_{\text{data}}(\mathbf{x}, \mathbf{y}, \mathbf{a}) = p_{\text{data}}(\mathbf{x}, \mathbf{a})\delta_{e_i(\mathbf{x})}(\mathbf{y})$, where $p_{\text{data}}(\mathbf{x}, \mathbf{a})$ is described by a dataset and $\delta_{e_i(\mathbf{x})}(\mathbf{y})$ is a Dirac conditional distribution enforcing that $\mathbf{y}$ is the CLIP embedding of $\mathbf{x}$. From now on all distributions denoted with $p_{\text{data}}$ should be understood as marginals and/or conditionals of this joint distribution, including our target distribution

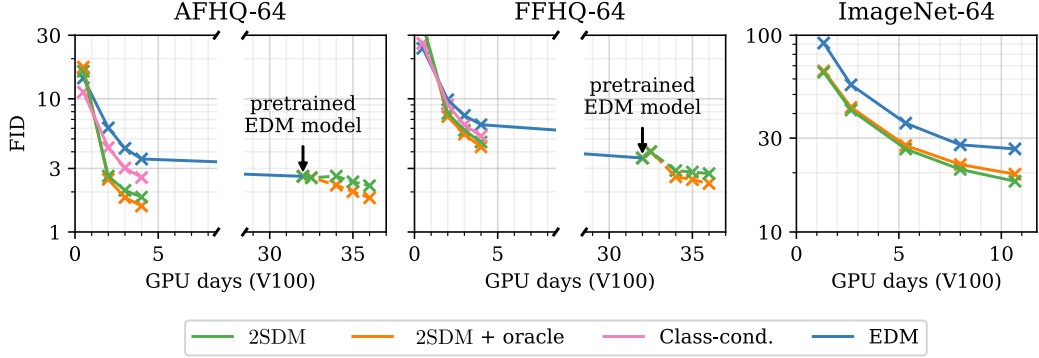

Figure 5: FID throughout training. We show results for each method trained from scratch and, on AFHQ and FFHQ, for finetuning a pretrained EDM model (which was trained for the equivalent of 32 GPU days). 2SDM quickly outperforms EDM when trained from scratch and quickly improves on the pretrained model when used for finetuning.

$p_{\text{data}}(\mathbf{x}|\mathbf{a})$. 2SDM approximates this target distribution as

$$p_{\text{data}}(\mathbf{x}|\mathbf{a}) = \mathbb{E}_{p_{\text{data}}(\mathbf{y}|\mathbf{a})}\left[p_{\text{data}}(\mathbf{x}|\mathbf{y},\mathbf{a})\right] \tag{2}$$

$$\approx \mathbb{E}_{p_\phi(\mathbf{y}|\mathbf{a})}\left[p_\theta(\mathbf{x}|\mathbf{y},\mathbf{a})\right] \tag{3}$$

where $p_\phi(\mathbf{y}|\mathbf{a})$ is a second DGM modeling the CLIP embeddings. We can sample from this distribution by sampling $\mathbf{y} \sim p_\phi(\cdot|\mathbf{a})$ and then leveraging the conditional image DGM to sample $\mathbf{x} \sim p_\theta(\cdot|\mathbf{y},\mathbf{a})$. We then return $\mathbf{x}$ and make no further use of $\mathbf{y}$. From now on we will call $p_\theta(\mathbf{x}|\mathbf{y},\mathbf{a})$ the *conditional image model* and $p_\phi(\mathbf{y}|\mathbf{a})$ the *auxiliary model*. In our experiments the auxiliary model uses a small architecture relative to the conditional image model and so adds little extra cost.[3]

**Auxiliary model** Our auxiliary model is a conditional DGM targeting $p_{\text{data}}(\mathbf{y}|\mathbf{a})$, where $\mathbf{y}$ is a 512-dimensional CLIP embedding. Following Eq. (1), we train it by minimizing

$$\mathbb{E}_{u(\sigma)p_\sigma(\mathbf{y}_\sigma|\mathbf{y},\sigma)p_{\text{data}}(\mathbf{y},\mathbf{a})}\left[\lambda(\sigma)||\mathbf{y} - \hat{\mathbf{y}}_\theta(\mathbf{y}_\sigma,\mathbf{a},\sigma)||^2\right]. \tag{4}$$

Analogously to Eq. (1), $\mathbf{y}_\sigma \sim p_\sigma(\cdot|\mathbf{y},\sigma)$ is a copy of the CLIP embedding $\mathbf{y}$ corrupted with Gaussian noise, and $u$ and $\mathbf{y}$ are the training distribution over noise standard deviations and weighting function respectively. We follow the architectural choice of Ramesh et al. (2022) and use a DGM with a transformer architecture. It takes as input a series of 512-dimensional input tokens: an embedding of $\sigma$; an embedding of $\mathbf{a}$ if this is not null; an embedding of $\mathbf{y}_\sigma$; and a learned query. These are passed through six transformer layers and then the output corresponding to the learned query token is used as the output. Like Ramesh et al. (2022), we parameterize the DGM to output an estimate of the denoised $\mathbf{a}$ instead of estimating the added noise as is more common in the diffusion literature. On AFHQ and FFHQ we find that data augmentation is helpful to prevent the auxiliary model overfitting. We perform augmentations (including rotation, flipping and color jitter) in image space and feed the augmented image through $e_i(\cdot)$ to obtain an augmented CLIP embedding. Following Karras et al. (2022), we pass a label describing the augmentation into the transformer as an additional input token so that we can condition on there being no augmentation at test-time.

**Conditional image model** Including the additional conditioning input $\mathbf{a}$, the conditional image model's training objective is

$$\mathbb{E}_{u(\sigma)p_\sigma(\mathbf{x}_\sigma|\mathbf{x},\sigma)p_{\text{data}}(\mathbf{x},\mathbf{y},\mathbf{a})}\left[\lambda(\sigma)||\mathbf{x} - \hat{\mathbf{x}}_\theta(\mathbf{x}_\sigma,\mathbf{y} \oplus \mathbf{a},\sigma)||^2\right]. \tag{5}$$

where $\mathbf{y} \oplus \mathbf{a}$ is the concatenation of $\mathbf{y}$ and $\mathbf{a}$ to form a single vector which the image model is conditioned on. We match our diffusion process hyperparameters, including $u$ and $\lambda$, to those of Karras et al. (2022), and also use their proposed Heun sampler. For AFHQ and FFHQ, we use the U-Net architecture originally proposed by Song et al. (2020). For ImageNet, we use the slightly larger

---

[3]For our ImageNet experiments, sampling from our auxiliary model takes 35ms per batch item. Sampling from our image model takes 862ms and so 2SDM has inference time only $4\%$ greater than our baselines.

Table 1: Comparison of 2SDM and EDM on a suite of metrics. Best performance for each metric and dataset is shown in bold. Higher is better for metrics marked ↑; lower is better for ↓. Results reported for EDM on FFHQ and AFHQ are computed with the pretrained checkpoints released by Karras et al. (2022). Results reported for 2SDM on FFHQ are with finetuning from this pretrained checkpoint. All others are trained from scratch.

| Dataset | Method | Inception Score ↑ | Precision ↑ | Recall ↑ | FID ↓ | sFID ↓ |
|---|---|---|---|---|---|---|
| AFHQ-64 | 2SDM | **10.00** | **0.844** | **0.619** | **1.56** | 13.7 |
| | EDM | 8.91 | 0.752 | 0.614 | 2.04 | 13.7 |
| FFHQ-64 | 2SDM | **3.47** | **0.721** | **0.697** | **2.32** | 4.98 |
| | EDM | 3.33 | 0.697 | 0.569 | 2.46 | **4.90** |
| Class-cond. ImageNet-64 | 2SDM | **17.3** | **0.541** | **0.573** | **17.4** | **4.63** |
| | EDM | 13.6 | 0.530 | 0.532 | 25.4 | 6.50 |
| Uncond. ImageNet-64 | 2SDM | **15.6** | **0.614** | **0.526** | **21.0** | **5.59** |
| | EDM | 11.3 | 0.523 | 0.524 | 35.1 | 9.14 |
| Class-cond. latent ImageNet-256 | 2SDM | **52.1** | **0.590** | 0.603 | **24.3** | **7.36** |
| | EDM | 40.4 | 0.532 | **0.610** | 34.2 | 9.59 |

U-Net architecture proposed by Dhariwal & Nichol (2021). We match the data augmentation scheme to be the same as that of Karras et al. (2022) on each dataset. There are established conditional variants of both architectures (Dhariwal & Nichol, 2021; Karras et al., 2022) that add a learned linear projection to the embedding of the noise standard deviation $\sigma$. We use the same technique to incorporate the concatenated conditioning inputs $\mathbf{y} \oplus \mathbf{a}$.

## 5 EXPERIMENTS

**Experimental setup and results overview** We perform experiments in five settings: unconditional AFHQ modeling at $64 \times 64$ resolution (Choi et al., 2020); unconditional FFHQ modeling at $64 \times 64$ resolution (Karras et al., 2018); unconditional ImageNet modeling at $64 \times 64$ resolution (Deng et al., 2009); class-conditional ImageNet modeling at $64 \times 64$ resolution; and finally class-conditional latent ImageNet modeling at $256 \times 256$ resolution, in which we train the diffusion models in the latent space of the pretrained VAE used by Stable Diffusion (Rombach et al., 2022). In every setting, we compare against EDM (Karras et al., 2022), a standard DGM directly modeling $p_{\text{data}}(\mathbf{x}|\mathbf{a})$, with an identical architecture to 2SDM. We match the training compute of our conditional image model with that of EDM in every case. The auxiliary model is trained for one day on a single V100 GPU so adds little additional cost. On AFHQ and FFHQ, we match the EDM parameters to those of Karras et al. (2022). On ImageNet-64, we have a smaller training budget and so decrease the batch size to 128 and the learning rate to $1 \times 10^{-4}$. For simplicity we match 2SDM to use the same learning rate and batch size.

For the first three of our listed settings, Fig. 5 reports the FID throughout the training of the conditional image diffusion model (or image DGM baseline).[4] In each case, the auxiliary model is trained for one day on one V100 GPU. We consider training the conditional image model from scratch (for up to 4 GPU days on AFHQ and FFHQ, or up to 11 GPU days on ImageNet-64), and see that it improves upon our EDM baseline for any training budgets over 1-2 GPU days. For AFHQ, this improvement is so substantial that 2SDM's FID after two GPU days is better than that of the pretrained EDM model released by Karras et al. (2022), which was trained for the equivalent of 32 V100 GPU days. In addition to training from scratch, on AFHQ and FFHQ we consider initializing 2SDM's training from the pretrained EDM checkpoints. To do so, we simply add a learnable linear projection of the CLIP embedding and initialize its

---

[4]Each FID in Fig. 5 is estimated using $20\,000$ images, each sampled with the SDE solver proposed by Karras et al. (2022) using 40 steps, $S_{\text{churn}} = 50$, $S_{\text{noise}} = 1.007$, and other parameters set to their default values. Our other reported FID scores use $50\,000$ samples, as is standard, and the same sampler hyperparameters.

weights to zero. We see that this allows for a fast and significant improvement in FID over the baseline in each case. We note, though, that training 2SDM from scratch for 4 GPU days outperforms 4 GPU days of finetuning on AFHQ and so recommend training 2SDM from scratch when sufficient compute is available.

Table 2: A comparison of FID with the state-of-the-art (SOTA) in bold. EDM (single seed) is our re-computation of the EDM's reported results using a single seed instead of taking the best of three.

| Dataset | AFHQ-64 | FFHQ-64 |
|---|---|---|
| PFGM++ (Xu et al., 2023) | – | 2.43 |
| EDM (Karras et al., 2022) | 1.96 | 2.39 |
| EDM (single seed) | 2.04 | 2.46 |
| EDM-G++ (Kim et al., 2022) | – | **1.77** |
| 2SDM | **1.56** | 2.31 |

Figure 4 also compares against "2SDM + oracle", which is a supposed upper bound on 2SDM's performance given by sampling a CLIP image embedding from an oracle (in practice, the dataset) and then using 2SDM's conditional image model to sample an image conditioned on it. It therefore describes the performance that 2SDM would achieve with a perfect auxiliary model. On AFHQ-64, 2SDM with an oracle achieves a FID $56\%$ lower than EDM. Without an oracle, 2SDM still achieves a FID $48\%$ lower than 2SDM. We therefore say that 2SDM yields an improvement $87\%$ as large as can be gleaned by using a purely conditional DGM. Similarly for FFHQ, 2SDM obtains an improvement $81\%$ as large as is possible with a purely conditional DGM.[5] We can therefore say that our cheaply-trained auxiliary model is good enough to allow us to capture the majority of the benefits of conditional generation for the unconditional generation task. Intriguingly, on ImageNet-64, 2SDM achieves better FID *without* an oracle. This suggests that imperfections in the distribution learned by the auxiliary model improve the visual quality of the generated images. We observed this trend consistently on ImageNet, and believe that characterizing exactly when and why it occurs is an intriguing direction for future work.

Finally, Fig. 5 also compares against "Class-cond", which is an ablation of 2SDM in which we replace the CLIP embedding $\mathbf{y}$ with a single discrete label obtained by K-means clustering of the CLIP embedding (as on the right of Fig. 4). For unconditional generation tasks, we can then replace our auxiliary model with a simple categorical distribution modeling $p_{\text{data}}(\mathbf{y}|\mathbf{a}) = p_{\text{data}}(\mathbf{y})$ similarly to Hu et al. (2022), simplifying the generative procedure. We see that this baseline is outperformed by 2SDM, justifying our choice to use a continuous $\mathbf{y}$.

We report our final FIDs on AFHQ and FFHQ alongside the state-of-the-art in Table 2. Despite our limited training budget, our results on AFHQ beat the state-of-the-art and our results on FFHQ come second to EDM-G++ (Kim et al., 2022), a potentially orthogonal approach to improving EDM.

**Latent diffusion on ImageNet-256** We combine 2SDM and the latent diffusion modeling framework (Rombach et al., 2022) on the ImageNet-256 dataset as follows. We take the pretrained Stable Diffusion VAE encoder and decoder released by Rombach et al. (2022). We feed a $256 \times 256 \times 3$ dataset image through the VAE encoder to create $64 \times 64 \times 4$ tensors, which we use as the training targets $\mathbf{x}$ for our conditional image model. The training targets for the CLIP embeddings $\mathbf{y}$ are created by embedding the $256 \times 256 \times 3$ images with the standard CLIP image embedder. We use the ImageNet class labels as additional inputs $\mathbf{a}$. At test time, we take $\mathbf{a}$ as an input; we then sample $\mathbf{y}$ given $\mathbf{a}$ from our auxiliary model; we then sample $\mathbf{x}$ given $\mathbf{y}$ and $\mathbf{a}$ from our conditional image model; we finally use the Stable Diffusion VAE decoder to produce an image given $\mathbf{x}$. Samples from this version of 2SDM, as well as our EDM baseline operating in the same latent space, are shown in Fig. 1. While the compute used for each (12 GPU days) is far from that of the state-of-the-art for this dataset, the samples from 2SDM are noticeably better, supporting the FID scores in Table 1.

**Diverse metrics** In Table 1 we show a comparison of 2SDM and EDM on a variety of metrics. The Inception Score (Salimans et al., 2016; Barratt & Sharma, 2018) measures the diversity of the output from an image classifier when run on sampled images. The Precision and Recall metrics (Kynkäänniemi et al., 2019) estimate, roughly speaking, the proportion of generated images that lie on the data manifold (Precision) and the proportion of dataset images that can be found within the

---

[5]See Table 3 for the FIDs used in these calculations.

manifold of generated images (Recall). The FID approximates the distance between the distribution of embeddings of dataset images and that of embeddings of generated images. The sFID is similar but uses an embedding with more spatial information. 2SDM outperforms EDM on 22 of the 25 metric-dataset combinations, and is outperformed on only 2.

**Comparison of relative improvements between tasks**    In terms of FID, and for the networks trained from scratch and matched for training compute, the percentage improvement of 2SDM over EDM is $48.2\%$ on AFHQ-64; $26.0\%$ on FFHQ-64; $31.5\%$ on class-conditional ImageNet-64; $40.2\%$ on unconditional ImageNet-64; and $28.9\%$ on class-conditional ImageNet-256. While these are all substantial improvements, we point out two comparisons in particular.

First, the gain from using 2SDM on unconditional ImageNet-64 ($40.2\%$) is greater than that on class-conditional modeling of the same dataset ($31.5\%$). This supports our argument that two-stage diffusion techniques like 2SDM can have even greater impact in unconditional (or lightly-conditional) generation than in the text-conditional (or strongly-conditional) setting in which they were originally introduced with unCLIP (Ramesh et al., 2022). Noting that the class label already contains some of the information stored in a CLIP embedding, this finding also fits with our discussion of the effects of conditioning in Section 3. The performance of an image model conditioned on just a class label (EDM on class-cond. ImageNet) should therefore be somewhere in between that of an unconditional image model (EDM on uncond. ImageNet) and that of a CLIP-conditional image model (2SDM, assuming the auxiliary model is good), leading to this finding.

Second, the $28.9\%$ improvement in performance for the latent diffusion model on ImageNet-256 is only slightly less than the $31.5\%$ improvement for pixel-space diffusion on class-conditional ImageNet-64. This confirms that 2SDM can be readily combined with the widely used latent diffusion framework.

**Inference speed**    Sampling from 2SDM does impose a small additional cost relative to EDM, since we must begin by sampling from the auxiliary model. In all experiments, when we use 40 diffusion steps, sampling from our auxiliary model takes 8.8s with batch size 256. This corresponds to 35ms per batch item. Our conditional image model and our EDM baseline use identical architecture (other than the projection of $\mathbf{y}$) and we could not detect a difference between their sampling timeswhich were 862ms per batch item on our ImageNet architecture and 789ms per batch item on our AFHQ and FFHQ architecture. This means that the increase in time due to using 2SDM instead of EDM is less than $4\%$. Furthermore, we can negate this increase by using two less sampling steps for the conditional image model. Table 5 in the appendix shows that this lets us make 2SDM faster than EDM with almost no effect on sample quality.

**Overfitting analysis**    We test for overfitting on AFHQ and FFHQ in the appendix through interpolation plots and nearest neighbour searches. We summarize these results in Fig. 6 by sampling 100 images from each method; computing the LPIPS distance of each one to every training set image and taking the minimum over all training set images; and then plotting the histogram of these minima. To create the black line, we use 100 training set images and take the minima over *non-zero* LPIPS distances to training set images to avoid them being reported as their own nearest neighbours. We can be confident that a method is overfitting if its curve is further to the left than the black curve. We see that both 2SDM and EDM ovefit slightly on AFHQ (which contains only $15\,000$ images) but no overfitting is visible on FFHQ

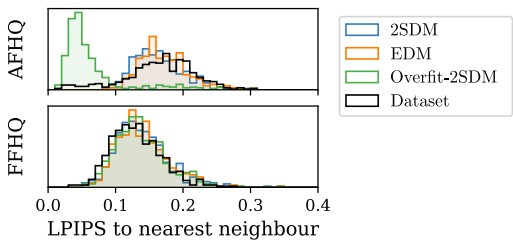

Figure 6: Distribution of LPIPS (Zhang et al., 2018) distances to the nearest neighbour in the training set for sampled images from EDM, 2SDM, and Overfit-2SDM. We see clear signs of overfitting for Overfit-2SDM on AFHQ but not for any other methods or datasets.

(which has $70\,000$ images). Seeing as these plots are similar for 2SDM and EDM, and given that ImageNet is a much larger dataset than AFHQ and FFHQ, we are confident that 2SDM's gains do not come from overfitting. We do, however, include another method, Overfit-2SDM, as a point of

interest and note of warning for future work on this topic. Overfit-2SDM is a variation of 2SDM in which we train the CLIP parameters jointly with the auxiliary and conditional image DGMs. It achieves state-of-the-art FID on AFHQ but, as we see in Fig. 6, only through near-total overfitting to the training set. See the appendix for more details.

## 6 RELATED WORK

**Intermediate variables in diffusion models**  Our work takes inspiration from Weilbach et al. (2022), who show improved performance in various approximate inference settings by modeling problem-specific auxiliary variables (like $\mathbf{y}$) in addition to the variables of interest ($\mathbf{x}$) and observed variables ($\mathbf{a}$). We apply these techniques to the image domain and incorporate pretrained CLIP embedders to obtain auxiliary variables.

**Latent diffusion**  2SDM also relates to methods which perform diffusion in a learned latent space (Rombach et al., 2022): our auxiliary model $p_\phi(\mathbf{y}|\mathbf{a})$ is analogous to a "prior" in a latent space and our conditional image model $p_\theta(\mathbf{x}|\mathbf{a}, \mathbf{y})$ to a "decoder" Such methods typically use a near-deterministic decoder and so their latent variables must summarize all information about the image. Our conditional DGM decoder, on the other hand, is a DGM that will function reasonably however little information is stored in $\mathbf{y}$. This means that 2SDM provides an additional degree of freedom in terms of what to store. Furthermore, as we showed in Section 5, 2SDM can be fruitfully combined with latent diffusion.

**Self-supervised representations**  Bao et al. (2022); Hu et al. (2022) both use self-supervised learning to obtain auxiliary variables and then training a diffusion model $p(\mathbf{x}|\mathbf{a})$. However, they do not model $\mathbf{a}$ and therefore are not able to sample $\mathbf{x}$ without an oracle that can provide $\mathbf{a}$. Their success when given an oracle, however, provides reason to believe that our approach is likely to yield benefits even if the embedder that produces $\mathbf{a}$ is obtained through self-supervised learning and without access to additional (or multi-modal) data as our CLIP embedder was trained with.

**Integrating additional data**  Our method can be understood as a means to leverage the "world knowledge" inside a CLIP embedder for improved performance on the image generation task. Another way in which additional knowledge, or data, could be leveraged is by training a multi-headed diffusion model which simultaneously approximates the score function and makes predictions of side information like class labels. Deja et al. (2023) propose a method for doing so but do not demonstrate improved performance on the unconditional generation task.

## 7 DISCUSSION AND CONCLUSION

We have demonstrated 2SDM, a variant of unCLIP for unconditional or lightly-conditional image generation, and argued that it has more benefits in this setting that in the text-conditional setting in which unCLIP was originally proposed. Therefore, even if the trend towards simple single-stage architectures continues for large-scale text-to-image models (Rombach et al., 2022; Chang et al., 2023; Hoogeboom et al., 2023), unCLIP-style approaches could offer large jumps in performance for lightly-conditional image generation tasks. 2SDM also holds promise for improving video generation. This is a domain for which CLIP could be readily applied, and being able to learn relationships in the relatively low-dimensional CLIP embedding space could significantly increase training throughput relative to working purely in pixel (or VAE embedding) space.

A massive unexplored design space remains. For pedagogical purposes we intentionally kept 2SDM simple, using known diffusion architectures and objectives. It is likely that optimizing these design choices for the lightly-conditional 2SDM use-case would improve performance. In addition, there are almost certainly more useful quantities that we could condition on than CLIP embeddings. Bao et al. (2022); Hu et al. (2022) have shown that self-supervised learning techniques provide a promising avenue for obtaining useful "latent" representations. Exactly the properties that an embedding should have to be beneficial for techiques like 2SDM is another open question that is ripe for future work to tackle. Such a line of work may also fix one limitation of 2SDM, namely that it relies on the availability of a pretrained CLIP embedder. While this is freely available for natural images, it could be a barrier to other applications. Improvements may also be gleaned by conditioning on multiple quantities, or "chaining" a series of conditional DGMs together. An alternative direction is to sim-

plify 2SDM's architecture by, for example, learning a single diffusion model over the joint space of **x** and **y** instead of generating them sequentially. We did not use classifier-free guidance (Ho & Salimans, 2022) in this work, which can improve visual fidelity at the cost of losing the mass-covering behavior that diffusion models are known for. Conditioning on the CLIP embedding with a high guidance scale could help to optimize for visual quality in future work.

## 8   ETHICS STATEMENT

Like much foundational research in modern generative modeling, this work carries risks like aiding the generation of deepfakes for dis- or misinformation campaigns. This leads to a second negative consequence: that trust in various forms of visual evidence, such as photographs, videos, and audio recordings, may no longer be possible. One avenue with which to address these consequences is research towards developing robust and effective methods for detecting and mitigating the harmful effects of deepfakes and synthetic media manipulation. Furthermore, increasing public awareness about the existence and potential impact of deepfakes can empower people to critically evaluate information and be more resilient to manipulation attempts. 2SDM has a potential risk on top of this: it leverages a publicly available "foundation model" in the form of a CLIP embedder to enhance the quality of generated content. Biases present in the foundation model may influence the outputs of 2SDM even if they are not present in the image dataset used for training. Stringent evaluation of foundation models may mitigate potential harms arising from this.

## 9   REPRODUCIBILITY STATEMENT

We release source code at https://anonymous.4open.science/r/2sdm. We will additionally release trained checkpoints on acceptance.

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

## A    ADDITIONAL RESULTS

Table 3 shows a breakdown of 2SDM's improvement over EDM by comparing how much it is improved when we do or do not have an oracle from which to sample CLIP embeddings.

In Table 4, we present our full suite of metrics evaluated on Overfit-2SDM as well as on 2SDM and our EDM baseline. Overfit-2SDM achieves extremely good scores on all metrics on AFHQ, where it was able to overfit excessively, illustrating that these metrics can be "gamed" by overfitting. On FFHQ, which is a larger dataset and which it appears to overfit to less, Overfit-2SDM does not achieve the best performance on any metric.

Table 5 compares sampling times for EDM, 2SDM, and a version of 2SDM that uses 2 fewer diffusion steps when sampling from the conditional image model. While 2SDM is slightly more expensive to sample from than EDM, taking 2 less sampling steps makes it cheaper with very little impact on sample quality.

Table 3: Final FID score for the models we train from scratch and a comparison of their improvements over EDM.

| Dataset | AFHQ | FFHQ | ImageNet |
|---|---|---|---|
| **y** | null | null | class label |
| EDM | 3.53 | 6.39 | 26.5 |
| 2SDM | 1.83 | 4.73 | 18.1 |
| 2SDM + oracle | 1.57 | 4.35 | 19.7 |
| Improv. w/ 2SDM | 48.2% | 26.0% | 31.5% |
| Improv. w/ oracle | 55.6% | 31.9% | 25.6% |
| $\frac{\text{Improv. w/ 2SDM}}{\text{Improv. w/ oracle}}$ | 86.6% | 81.3% | 123% |

Table 4: Comparison of Overfit-2SDM (O-2SDM) with 2SDM and EDM, following Table 1

| | AFHQ | | | FFHQ | | |
|---|---|---|---|---|---|---|
| | 2SDM | O-2SDM | EDM | 2SDM | O-2SDM | EDM |
| Inception Score $\uparrow$ | 10.00 | **10.70** | 8.91 | **3.47** | 3.37 | 3.33 |
| Precision $\uparrow$ | 0.844 | **0.977** | 0.752 | **0.721** | 0.695 | 0.697 |
| Recall $\uparrow$ | 0.619 | **0.869** | 0.614 | **0.697** | 0.552 | 0.569 |
| FID $\downarrow$ | 1.56 | **1.16** | 2.04 | **2.32** | 3.33 | 2.46 |
| sFID $\downarrow$ | 13.7 | **10.6** | 13.7 | 4.98 | 5.37 | **4.90** |

Table 5: Comparison of FID and sampling time for our EDM baseline, our proposed method 2SDM, and a variation of 2SDM where we use fewer sampling steps such that its sampling time is less than that of EDM. To account for batching, we measure the time to sample a batch of size 256 on a V100 GPU, and then divide it by 256. We see that (a) 2SDM outperforms EDM at little additional cost, and (b) we can reduce the number of sampling steps used by 2SDM such that it is both faster than and has better image quality than our baseline.

| | EDM | 2SDM | 2SDM-"less steps" |
|---|---|---|---|
| Sampling time | 789 ms | 824 ms | 784 ms |
| FID on AFHQ (trained from scratch) | 2.04 | 1.56 | 1.58 |
| FID on FFHQ (init. from pretrained EDM) | 2.46 | 2.32 | 2.32 |

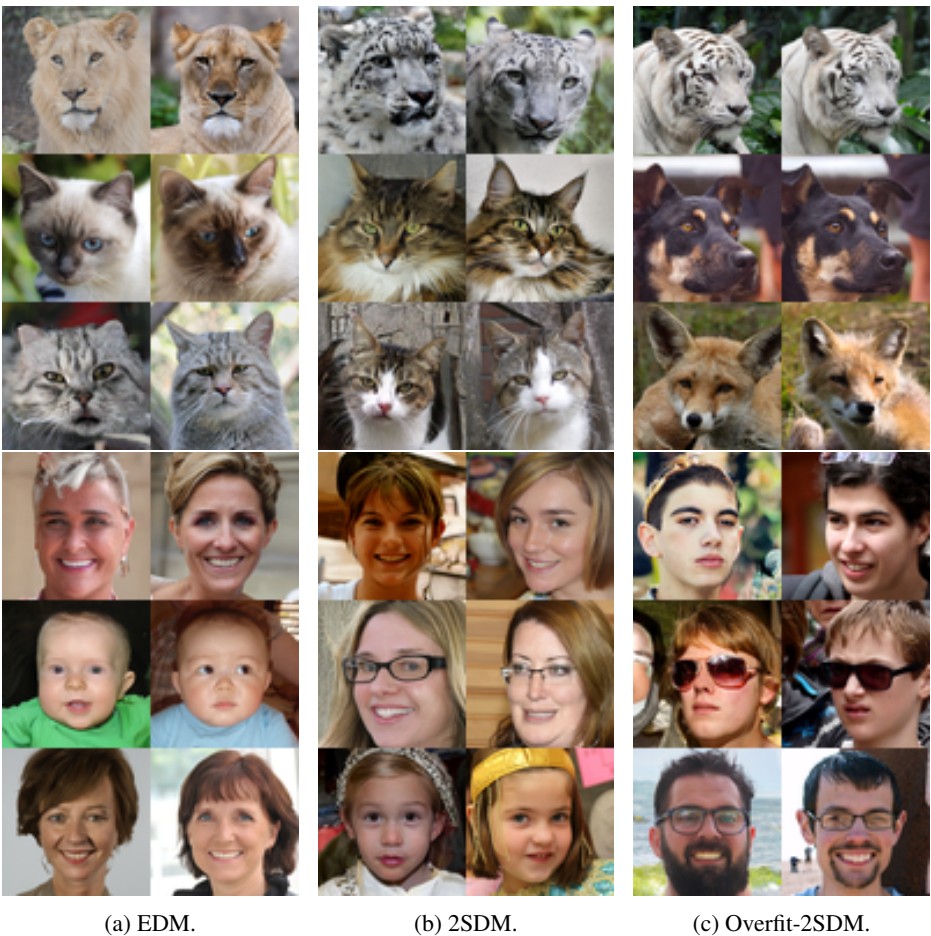

(a) EDM.        (b) 2SDM.        (c) Overfit-2SDM.

Figure 7: Samples (left of each pair of columns) and their nearest neighbours from the training set (right of each pair of columns) for each method. The top three rows use AFHQ; the bottom three use FFHQ. On AFHQ, Overfit-2SDM near-perfectly reconstructs training images, while both EDM and 2SDM both generate images that are meaningfully different to any training images. We show nearest neighbours for more samples in the appendix, as well as nearest neighbours computed in more embedding spaces (pixel space and LPIPS space).

## B  TRAINING THE EMBEDDER USING AN AUTOENCODER-STYLE OBJECTIVE

In this section we describe an approach to learning an embedder by maximizing an autoencoder-style objective, instead of using a pretrained CLIP embedder. We include this as an interesting negative result, as we found that doing so led to severe overfitting. For this experiment, we initialize the embedder from scratch and train it jointly with both diffusion models to minimize a combined diffusion loss, specifically the sum of Eq. (4) and Eq. (5). The combined diffusion loss can be interpreted as a weighted variant of a lower-bound on the likelihood of $(\mathbf{x}, \mathbf{y})$ pairs given $\mathbf{a}$. Even though this is ill-posed when $\mathbf{y}$ is a learned function of $\mathbf{x}$, Silvestri et al. (2022) show that it can be a good heuristic for maximizing the marginal likelihood of $\mathbf{x}$ given $\mathbf{a}$, which is ideal as we wish to target $p_{\text{data}}(\mathbf{x}|\mathbf{a})$.

Concretely, for this ablation we start with a randomly initialized ViT-B/32 network as the embedder $e_i$, matching the architecture of the CLIP model used by 2SDM. We then train the two diffusion models using the combined diffusion loss but additionally backpropagate gradients of this loss through $\mathbf{y} = e_i(\mathbf{x})$ to estimate gradients with respect to the embedder's parameters. Since the gradients become inter-dependent between the auxiliary model, the conditional image model, and the embedder, doing so requires that we train them all simultaneously. Doing so without excessive memory

usage requires gradient accumulation, in which we first compute and differentiate Eq. (4), and then compute and differentiate Eq. (5). Since the embedder parameters depend on both, we must wait until after having accumulated both sets of gradients to take an optimizer step. For reasons that will soon become clear, we call the resulting model "Overfit-2SDM."

Overfit-2SDM achieves a striking FID score of 1.16 on AFHQ, outperforming both 2SDM and the previous state-of-the-art by a wide margin. We also improve upon 2SDM and EDM in terms of all other metrics listed in Table 1, pointing to a weakness of the current evaluation metrics for deep generative models. The issue is clear in the nearest neighbour plots of Fig. 6. Samples from Overfit-2SDM are near-perfect reconstructions of the AFHQ training images. It is able to memorize them so well by simply mapping each to a distinct location in the 512-dimensional embedding space, as we show with interpolation plots in the appendix. To summarize the results of such nearest neighbours-analysis over many sampled images, Fig. 6 displays the distribution over LPIPS distances to nearest neighbours in the training set for 500 sampled images. As a "target", we plot in black a histogram of the distribution of such distances for training set images themselves. Assuming that the training set is sampled i.i.d. from a data-generating distribution $p_{\text{data}}(\mathbf{x})$, we can interpret this as the distribution of distances to training set images for samples from $p_{\text{data}}(\mathbf{x})$. Then, given that a perfectly well-fit generative model should produce samples from $p_{\text{data}}(\mathbf{x})$, the disparity between this distribution and the distribution for samples from each method gives us a way of evaluating each method. We see that the histograms match well for EDM and 2SDM, suggesting that neither is overfitting severely. For Overfit-2SDM on AFHQ, however, the sampled images are usually much closer to their nearest neighbours than expected, demonstrating overfitting.

On FFHQ (which is a larger dataset with $70\,000$ images vs. $15\,000$ for AFHQ) we do not see such overfitting from Overfit-2SDM but we do see worse FID scores, with Overfit-2SDM converging to a FID of 3.33, worse that the score reach by EDM. This tells us that learning the encoder in this manner does not lead to it having a beneficial effect for conditional generation unless the model is able to completely overfit to the training data. The beneficial effect of 2SDM therefore appears to come from a property of the pretrained CLIP embeddings that is not learnable through a simple reconstruction loss.

## C  INTERPOLATIONS

In Figs. 8 and 9 we show interpolations by Overfit-2SDM and 2SDM (respectively) between sampled images. These are reasonably smooth both when moving from left to right (varying the image-space noise) and when moving from top to bottom (varying the noise used to sample the CLIP embedding), suggesting that neither method overfits substantially on the FFHQ dataset, and confirming that the state-of-the-art FID scores achieved by 2SDM on FFHQ can not be explained away by overfitting.

We show interpolation between images sampled by Overfit-2SDM in Fig. 10. This confirms that overfit occurs in the model of the CLIP embedding rather than in the model of images given the CLIP embedding, since the path between images is smooth when moving from left to right (varying the noise applied in the image diffusion model) but jumps between concentrated peaks when moving from top to bottom (varying the noise used to sample the CLIP embedding).

## D  NEAREST NEIGHBOURS

In Figs. 11 to 13 we show the nearest neighbours for samples from various methods, computed in CLIP, LPIPS, and pixel space respectively. The nearest neighbours are more visually close when computed in CLIP or LPIPS space but, in all cases, the results are qualitatively similar. Overfit-2SDM overfits significantly, while EDM and 2SDM do not exhibit such noticeable overfitting.

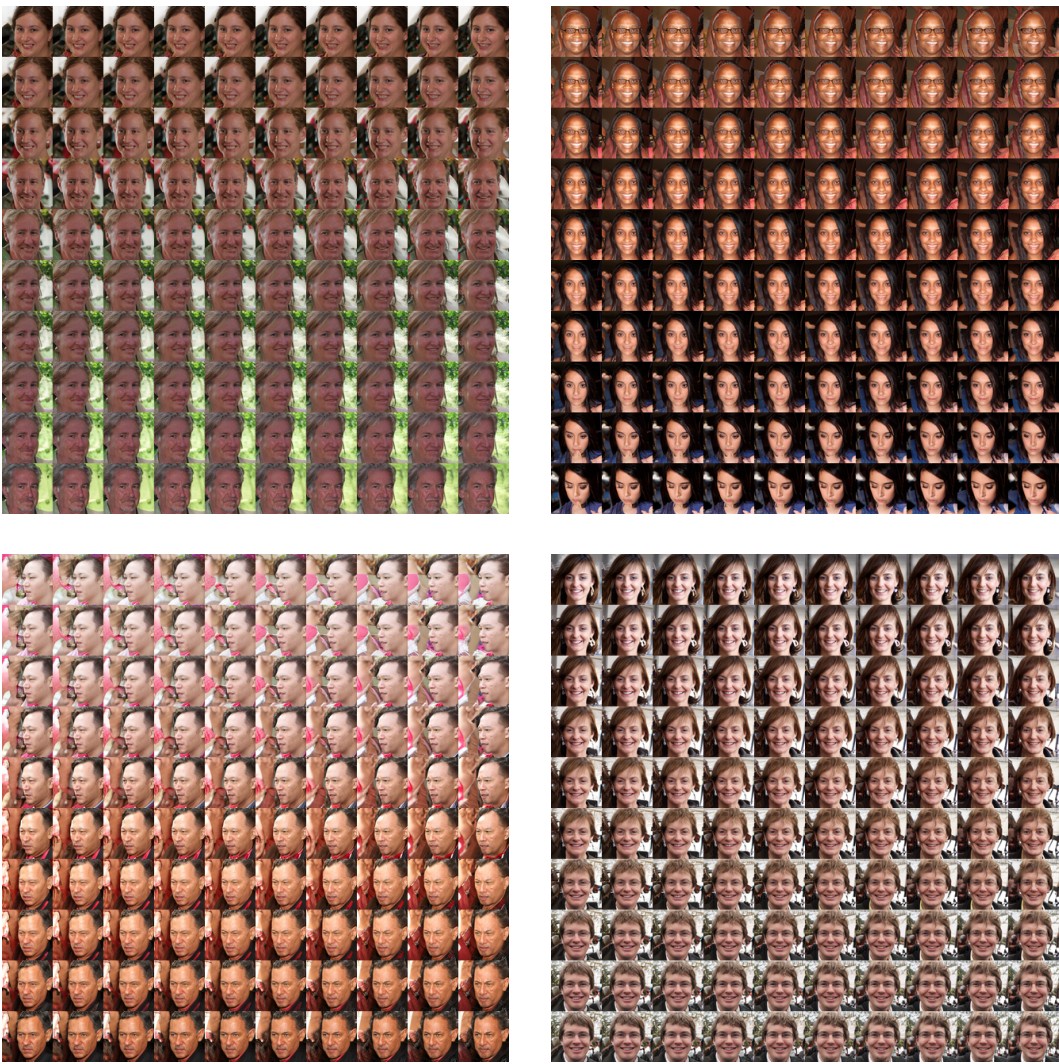

Figure 8: Overfit-2SDM interpolations between sampled FFHQ images. Within each row, the same sampled CLIP embedding is used. Within each column, the same image-space noise is used to sample an image given a CLIP embedding. We randomly sample the CLIP embeddings and noise for the top-left and bottom-right images, and use spherical interpolation between them to produce all other images.

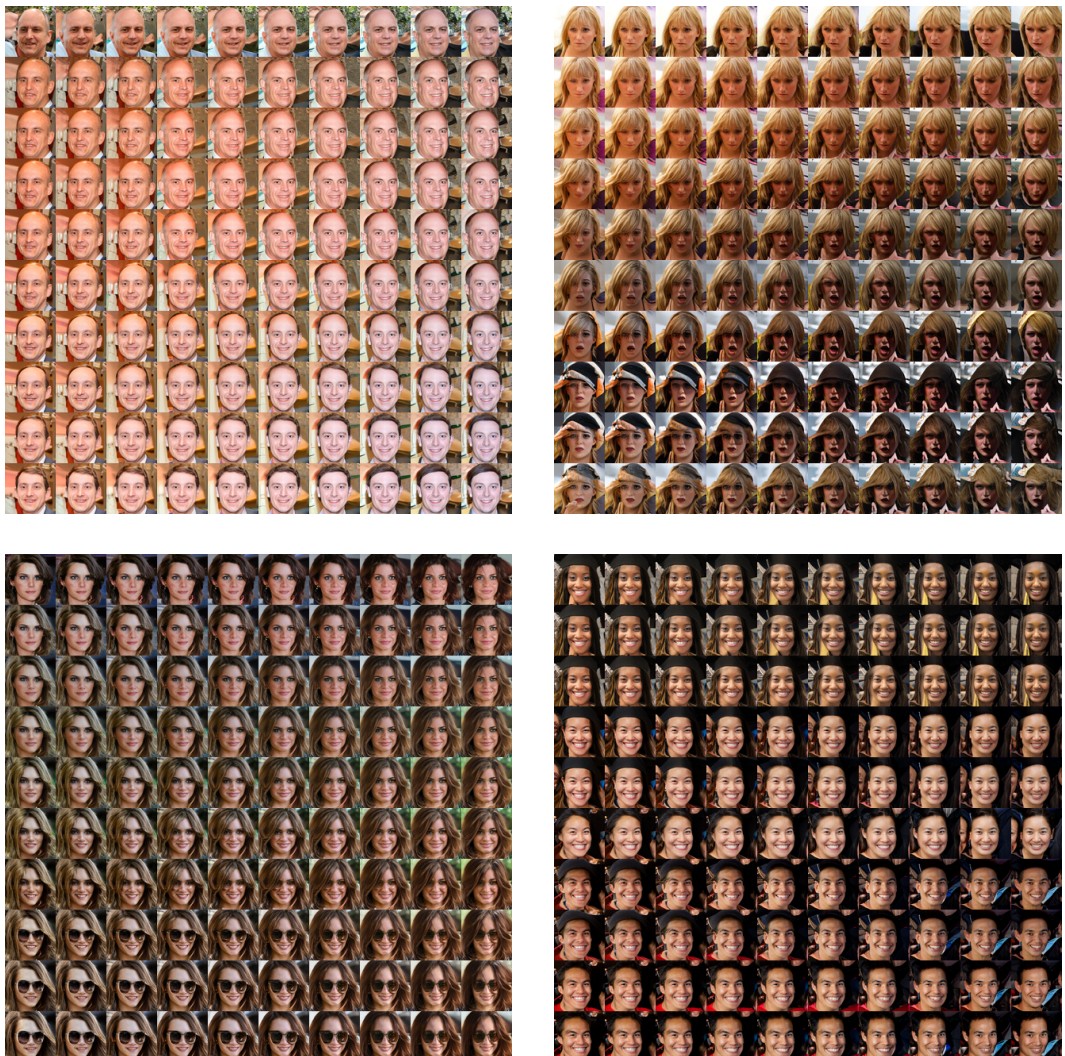

Figure 9: 2SDM interpolations between sampled FFHQ images. Within each row, the same sampled CLIP embedding is used. Within each column, the same image-space noise is used to sample an image given a CLIP embedding. We randomly sample the CLIP embeddings and noise for the top-left and bottom-right images, and use spherical interpolation between them to produce all other images.

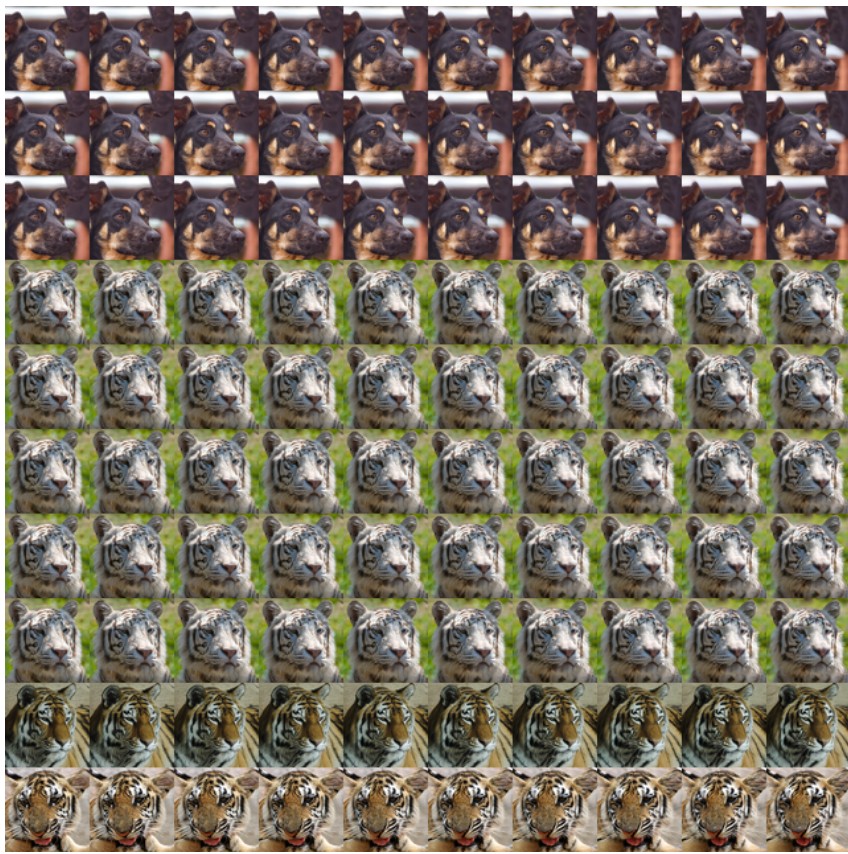

Figure 10: Interpolation between different sampled images by Overfit-2SDM. Within each row, the same sampled CLIP embedding is used. Within each column, the same image-space noise is used to sample an image given a CLIP embedding. We randomly sample the CLIP embeddings and noise for the top-left and bottom-right images, and use spherical interpolation between them to produce all other images.

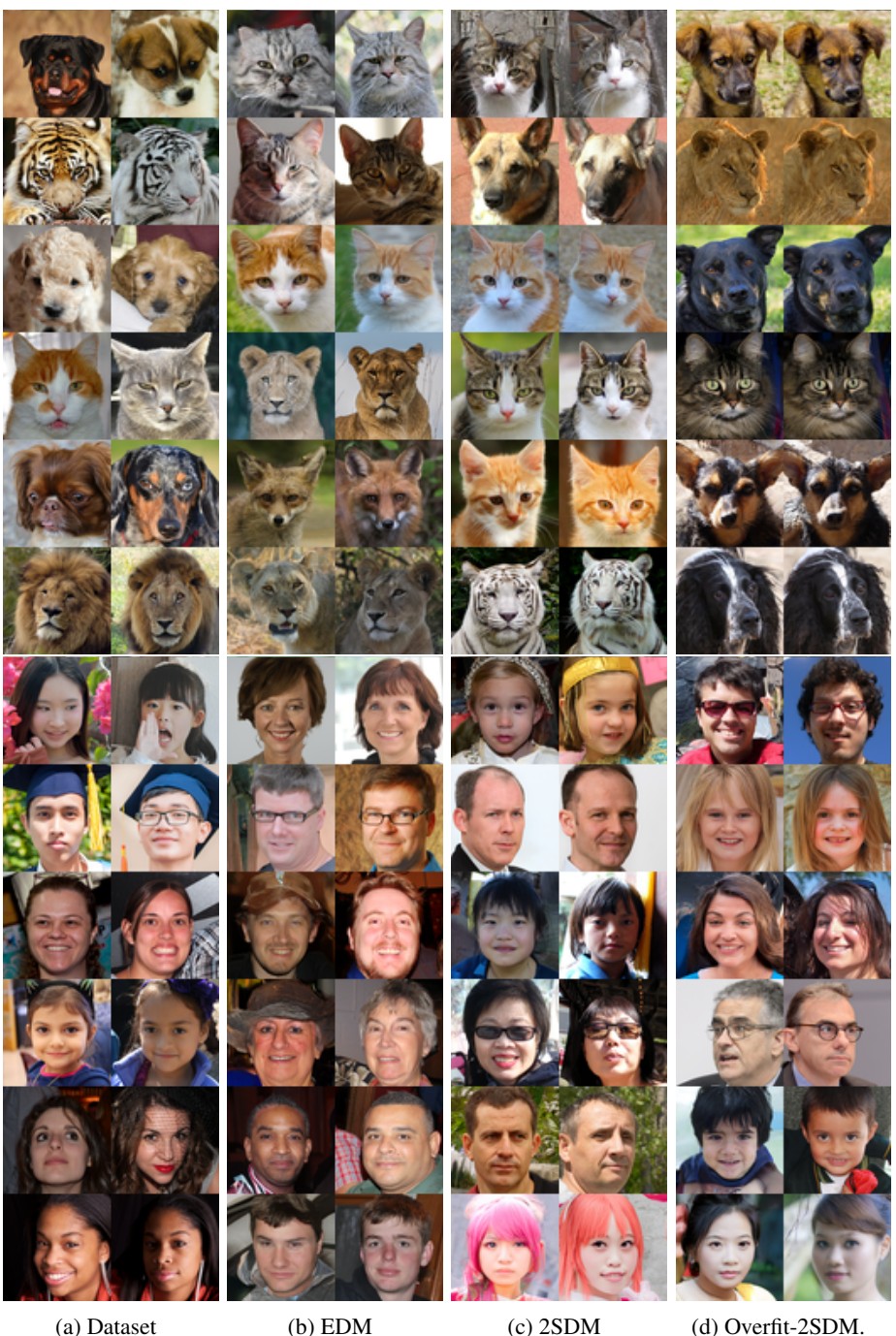

(a) Dataset          (b) EDM          (c) 2SDM          (d) Overfit-2SDM.

Figure 11: CLIP-space nearest neighbours, similar to Fig. 6. The images on the right of each column are the nearest training set image to the image on their left. As a baseline, column (a) shows the nearest neighbours for training set images (computed by explicitly preventing the training set images themselves being chosen as their nearest neighbour).

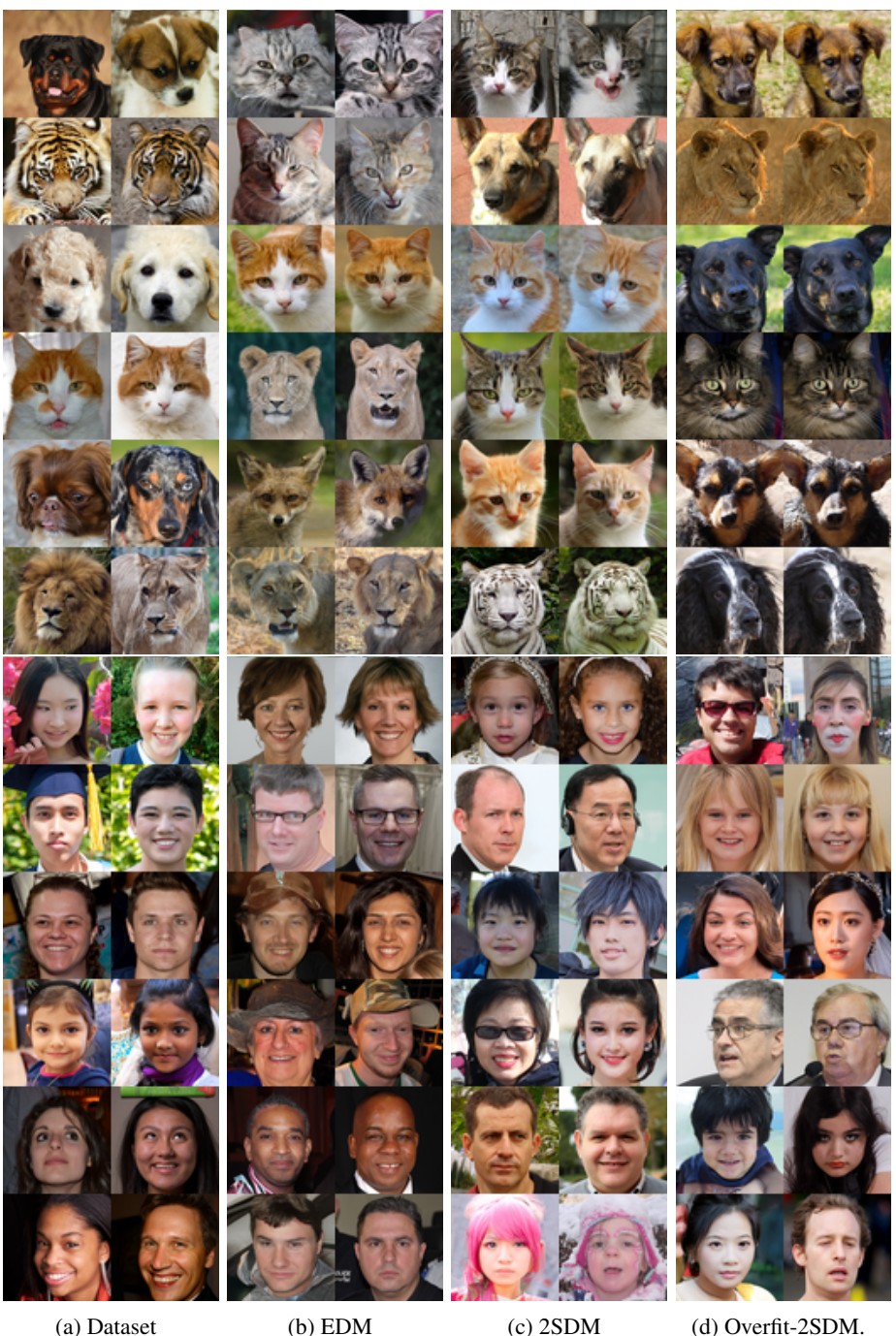

|  (a) Dataset | (b) EDM | (c) 2SDM | (d) Overfit-2SDM. |

Figure 12: LPIPS-space nearest neighbours. The images on the right of each column are the nearest training set image to the image on their left. As a baseline, column (a) shows the nearest neighbours for training set images (computed by explicitly preventing the training set images themselves being chosen as their nearest neighbour).

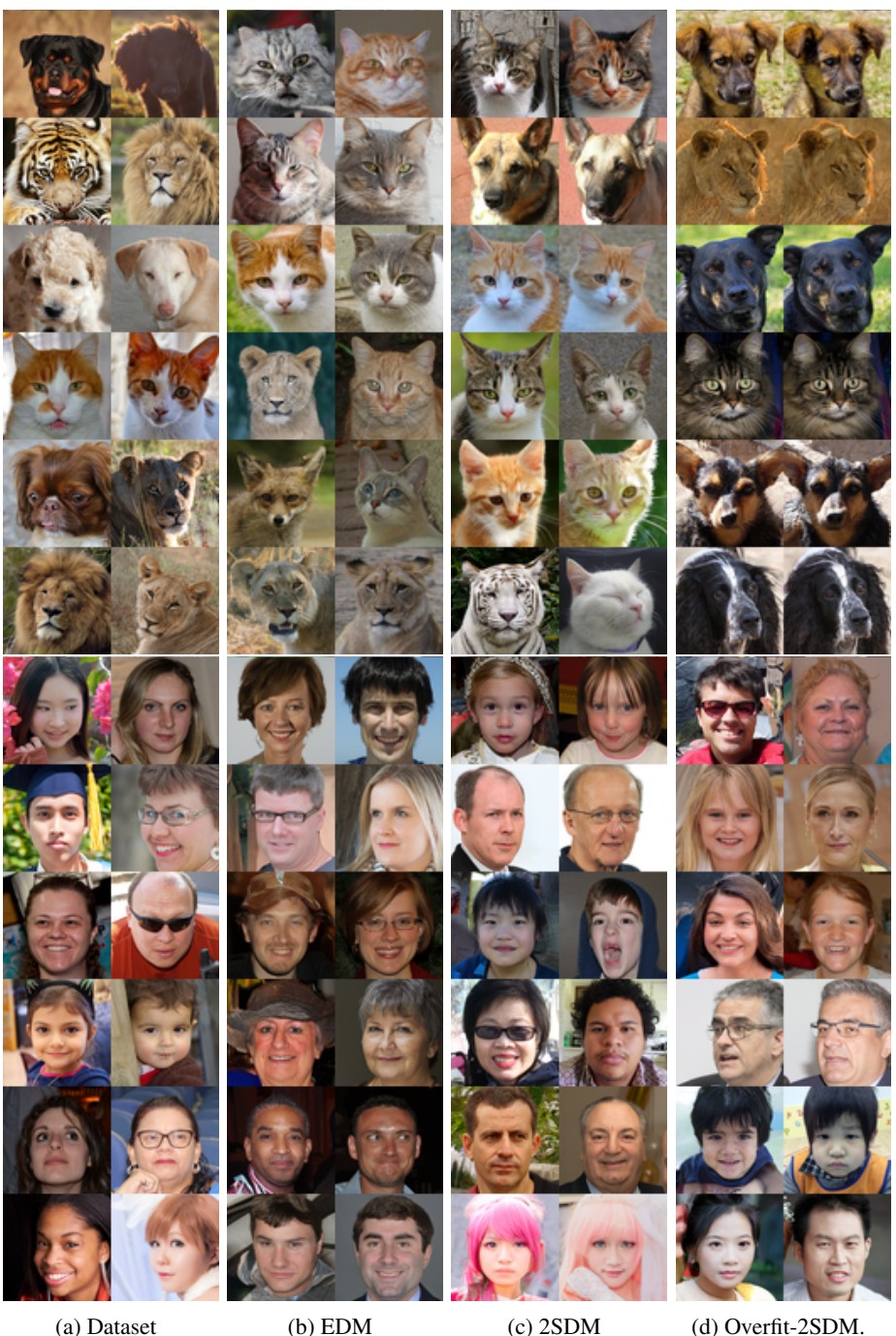

(a) Dataset          (b) EDM          (c) 2SDM          (d) Overfit-2SDM.

Figure 13: Pixel-space nearest neighbours. The images on the right of each column are the nearest training set image to the image on their left. As a baseline, column (a) shows the nearest neighbours for training set images (computed by explicitly preventing the training set images themselves being chosen as their nearest neighbour).

## E   EXPERIMENTAL DETAILS

**Compute**   The total compute spent on this project, including unreported preliminary runs, was approximately 1 GPU year. We used a mixture of 16GB Tesla V100 and 40GB NVIDIA A100 GPUs. Our training times for 2SDM and our EDM baselines are shown in Fig. 5. Overfit-2SDM was slower to converge and we trained it for roughly 24 GPU days on both AFHQ and FFHQ.

## F   LICENCES

**Datasets:**

- FFHQ (Karras et al., 2018): Creative Commons BY-NC-SA 4.0 license
- AFHQv2 (Choi et al., 2020): Creative Commons BY-NC 4.0 license
- ImageNet (Deng et al., 2009): The license status is unclear

**Pre-trained models:**

- EDM models by Karras et al. (2022): Creative Commons BY-NC-SA 4.0 license

