# OpenReview forum: "Two-Stage Diffusion Models: Better Image Synthesis by Explicitly Modeling Semantics"
_ICLR.cc/2024/Conference — ICLR 2024 Conference Withdrawn Submission_

### Official Review · Reviewer_c6YU · 2023-10-31

**Soundness:** 2 fair
**Presentation:** 3 good
**Contribution:** 2 fair
**Rating:** 3
**Confidence:** 3

**Summary:**

The paper revisits the unCLIP paradigm proposed by Ramesh et al., 2022, which consists of two cascaded diffusion models, one trained on CLIP latent text embeddings and another one mapping from latent text embeddings to the image space. Unlike unCLIP, the paper proposes to train the latent diffusion models on CLIP image embeddings rather than text embeddings, which enables unconditional image generation. The proposed model is evaluated on different variants of AFHQ, FFHQ, and ImageNet, and is compared to EDM (Karras et al. 2022) among other baselines.

**Strengths:**

Improving unconditional image generation models is an active research area, and lags substantially behind class/text conditional generation. Making progress in this area is important. Further, techniques relying on multi-stage modeling/latent modeling like the proposed one have proven effective in making diffusion models more efficient.

**Weaknesses:**

I see two main weaknesses:
1. The lack of novelty. The proposed method is very similar to unCLIP.
2. The method is arguably more complicated than (Hu et al., 2022) which simply clusters image embeddings obtained by a self-supervised representation and uses the cluster indices as conditioning signal. While the paper compares to this approach on AFHQ/FFHQ, I’m not fully convinced that the proposed method is superior. I would expect a comparison on ImageNet to be convinced that the additional complexity of the proposed approach is justified, since Hu et al. get similar improvements.

Given these two points, I’m leaning towards rejecting this paper.


Minor points:
- Typo page 2 bottom “that the all images”
- I found the terms lightly/strongly conditional somewhat confusing. Maybe it would be simpler to just use class/text conditional?

**Questions:**

- Do the authors have any explanation why the 2SDM outperforms 2SDM with oracle in Figure 5 right?
- Did the authors consider any other image embeddings besides CLIP? For example DINO might be better aligned with ImageNet. Also it would be interesting to see how well the first diffusion model can learn the embedding, and how this affects the quality of the end-to-end model.

---

> ### Author Response · Authors · 2023-11-23
>
> Thank you for your review and very helpful comments. We will withdraw this submission, but will also reply to your comments here for posterity.
>
> *Comparison to Hu et al. on ImageNet., and comparison with other image embeddings* - Thank you for these suggestions! We had not tried using other embeddings like DINO but expect that they would work well based on the results in [Self-Guided Diffusion Models](https://arxiv.org/abs/2210.06462). Unfortunately we (due to a bug in our codebase introduced while running new experiments) have not been able to present them by the end of the discussion period.
>
> *Typo* - Fixed, thanks for catching this!

---

### Official Review · Reviewer_5Hdf · 2023-11-01

**Soundness:** 3 good
**Presentation:** 2 fair
**Contribution:** 2 fair
**Rating:** 6
**Confidence:** 3

**Summary:**

The paper proposes an unconditional image generation pipeline which is split into two parts: first, a model generates a random condition (in this case a CLIP image embedding), then, a second model is conditioned on this condition to generate the actual image. Compared to baseline unconditional single-stage models, the new approach performs better while leading only to a small overhead in training and sampling cost.

**Strengths:**

The approach tackles unconditional image generation, and improvements in that area could potentially also translate to conditional generation pipelines.

The approach of splitting the unconditional generation into two parts seems novel and the results indicate that this does indeed lead to improvements, at least on the relatively small datasets and image resolutions that it was tested on.

**Weaknesses:**

While the approach seems to lead to improved performance it's not clear to me why this is the case and there is only very little analysis around this.
Is it that unconditional sampling of CLIP image embeddings is somehow important or easier than sampling an image directly? Or is it the two-stage pipeline itself that is the important part? Could the condition generation and subsequent image generation be done in a single pipeline with end-to-end training? What exactly is the interaction between the first and second stage models?

**Questions:**

How well do you think this would work for more complicated domains and datasets?
How do you think this approach could benefit/improve conditional generation pipelines such as text-to-image?
How well do you think this would work with more specific conditions in the first stage (e.g., depth maps, edge maps, etc)?

---

> ### Author Response · Authors · 2023-11-23
>
> Thank you for your time and very helpful comments. We have decided to withdraw our submission but will also reply to your comments here for the record.
>
>
> *“Is it that unconditional sampling of CLIP image embeddings is somehow important or easier than sampling an image directly”* - Yes, we believe that the distribution over CLIP embeddings of natural images is much simpler and easier to model than the distribution over natural images themselves. One reason for this is that they are lower-dimensional than natural images. Another, we suspect, is that CLIP embedding space is much smoother than pixel space. See e.g. how CLIP embeddings can be reasonably manipulated with simple "vector arithmetic" in Section 3.3 of [DALL-E 2](https://arxiv.org/abs/2204.06125).
>
> *Could the condition generation and subsequent image generation be done in a single pipeline with end-to-end training?* - We tried training the encoder/CLIP embedder jointly with the auxiliary model and conditional image model in Appendix B. Our results were substantially worse than when we have a pretrained CLIP embedder, and we observed severe overfitting on AFHQ. We suspect that the pretrained features from CLIP, and their alignment to features noticeable to humans, are a large part of why 2SDM works well.
>
> *What exactly is the interaction between the first and second stage models?* - These models can be trained separately and independently. As long as they are both trained with the same CLIP embedder, the outputs from the first stage model can be fed into the second stage model at inference time to produce coherent output images.
>
> *How well do you think this would work for more complicated domains and datasets?* - The benefit of 2SDM over our baselines is related to how much additional information the CLIP embedding $\mathbf{y}$ can provide over $\mathbf{a}$. In the unconditional case, 2SDM’s advantage should grow with the diversity of a dataset, since e.g. having a CLIP embedding describing that an image is of a man’s face is much more informative in a dataset like ImageNet where few of the images are of faces than it would be in FFHQ, where roughly 50% of the images are men’s faces. This is supported by our greater reduction in FID on Uncond. ImageNet-64 than on FFHQ-64. On the other hand, 2SDM’s advantage will decrease as the conditioning information $\mathbf{a}$ becomes more complex, since this will reduce the amount of information that $\mathbf{y}$ provides and $\mathbf{a}$ doesn’t. This is supported by our greater reduction in FID on Uncond. ImageNet-64 than on Class-cond. ImageNet-64.
>
> *more specific conditions e.g., depth maps* - We think it’s likely that conditioning on this type of information can certainly improve generation quality. Training an auxiliary model which outputs depth maps is likely to be more difficult, though, than one which outputs CLIP embeddings, and so it is not clear to us whether a two-stage model like 2SDM will outperform a one-stage model if $\mathbf{y}$ is of this form.

---

### Official Review · Reviewer_WB76 · 2023-11-01

**Soundness:** 2 fair
**Presentation:** 1 poor
**Contribution:** 2 fair
**Rating:** 3
**Confidence:** 3

**Summary:**

This paper introduces a two-stage approach, 2SDM, for sampling from diffusion models. The goal is to improve the performance of unconditional generation, which has a gap in performance compared to conditional generation. In the first stage, an auxiliary diffusion model is used to generate an embedding, which is subsequently used in the second stage by a conditional diffusion model to synthesize an image. The authors demonstrate that 2SDM yields better performance across almost all experiments in terms of quality and diversity with little to no increase in sampling speed.

**Strengths:**

- The proposed two-stage approach is straightforward, extending UnCLIP to the unconditional setting.
- The authors provide some context and insight into what it means for conditional generation to be better than unconditional generation, and why this may be the case.
- Experiments demonstrate superior performance over the baselines with negligible impact on the sampling speed.

**Weaknesses:**

While the method is straightforward, the paper is a bit difficult to understand overall. The finer details are unclear. For example:
- It is unclear what the authors mean when they mention "discarding" the conditional embedding y after sampling.
- The details about the auxiliary model in Section 4 are unclear. For example, what is a_\sigma? Maybe reiterating some of the variables in Equation 4 would be helpful, too.
- In the results overview of Section 5, the authors describe that Figure 4 (which seems to actually be referring to Figure 5) "compares against 'Class-cond', which is an ablation of 2SDM that applies to unconditional tasks". Given the label "Class-cond" it seems more intuitive that this would refer to the "lightly-conditional" task instead.

**Questions:**

- For explicit clarification, are the two models (auxiliary and conditional image) trained sequentially?
- The authors mention that they did not use classifier-free guidance in their results, which is common practice for diffusion sampling. It would be helpful to get some sense of how it affects the quality of the outputs.
- The experimental results are compelling and the method is straightforward, but the paper could greatly benefit from clearer communication of the proposed ideas and details.

---

> ### Author Response · Authors · 2023-11-23
>
> Thank you for your very helpful comments. We have decided to withdraw this submission but will also reply to your comments here for posterity.
>
> *"discarding" the conditional embedding y after sampling* - Thanks for pointing out that this wasn’t clear. We were simply meaning to make explicit that $\mathbf{y}$ isn’t used again after $\mathbf{x}$ is sampled. We have rewritten this in Section 4.
>
> *details about the auxiliary model* - Apologies; a typo made this section confusing: $\mathbf\{a\}\_\sigma$ should have been $\mathbf{y}_\sigma$. We have now fixed it and added an extra explanatory sentence.
>
> *'Class-cond' baseline description* - The ``class-cond.’’ method is a version of 2SDM in which $\mathbf{y}$ is a discrete class label instead of a continuous CLIP embedding. We say it is particularly applicable to the unconditional setting because then $\mathbf{y}$ can be reasonably sampled from the empirical distribution over $\mathbf{y}$ represented by the training data without training any auxiliary model. We have now made this paragraph clearer.
>
>  *are the two models (auxiliary and conditional image) trained sequentially?* - The two models can be trained sequentially or concurrently, since the training of the conditional image model does not rely on already having an auxiliary model, and vice versa.
>
> *classifier-free guidance* - Thank you for this suggestion! We unfortunately haven’t been able to obtain these results before the end of the discussion period but in a future revision will add results in which we vary the classifier-free guidance scales for each of our baseline image model, 2SDM’s auxiliary model, and 2SDM’s conditional image model.